# Predictors of Length of Stay, Mortality and Rehospitalization in COPD Patients: A Retrospective Cohort Study

**DOI:** 10.3390/jcm12165322

**Published:** 2023-08-16

**Authors:** Giorgia Lüthi-Corridori, Maria Boesing, Nicola Ottensarendt, Anne Barbara Leuppi-Taegtmeyer, Philipp Schuetz, Joerg Daniel Leuppi

**Affiliations:** 1University Center of Internal Medicine, Cantonal Hospital Baselland, 4410 Liestal, Switzerland; giorgia.luethi-corridori@ksbl.ch (G.L.-C.);; 2Faculty of Medicine, University of Basel, 4056 Basel, Switzerland; 3Department of Patient Safety, Medical Directorate, University Hospital Basel, 4056 Basel, Switzerland; 4Cantonal Hospital Aarau, University Department of Medicine, 5001 Aarau, Switzerland

**Keywords:** AECOPD, LOHS, rehospitalization, mortality

## Abstract

Chronic obstructive pulmonary disease (COPD) is a highly prevalent chronic lung disease that has a significant impact on individuals and healthcare systems worldwide. This study aimed to identify factors that predict the length of a hospital stay (LOHS), one-year mortality, and rehospitalization within 6 months in patients admitted for acute exacerbation of COPD (AECOPD). A retrospective cohort study was conducted using data from 170 patients admitted to a district general hospital in Switzerland between January 2019 and February 2020. Sociodemographic and health-related variables measured at admission were analyzed as potential predictors. Multivariable zero-truncated negative binomial and logistic regression analyses were performed to assess the risk factors for LOHS (primary endpoint), mortality, and rehospitalization. The results show that an indication for oxygen supplementation was the only significant predictor of LOHS. In the logistic regression analysis, older age, COPD severity stages GOLD III and IV, active cancer and arrhythmias were associated with higher mortality, whereas rehabilitation after discharge was associated with lower mortality. There were no significant associations regarding rehospitalization. This study identified routinely available predictors for LOHS and mortality, which may further advance our understanding of AECOPD and thereby improve patient management, discharge planning, and hospital costs. The protective effect of rehabilitation after hospitalization regarding lower mortality warrants further confirmation and may improve the comprehensive management of patients with AECOPD.

## 1. Introduction

Chronic obstructive pulmonary disease (COPD) is “a common preventable and treatable chronic lung disease” that is the third leading cause of death worldwide [1]. In Switzerland, approximately 400,000 people suffer from COPD [2]. COPD imposes a significant medical and financial strain on patients and healthcare systems, resulting in a diminished quality of life, reduced life expectancy, and substantial healthcare expenditures globally [3,4]. 

In addition, COPD patients often are affected by other life-challenging comorbidities, such as ischemic heart disease, osteoporosis, musculoskeletal disorders, lung cancer, and depression or anxiety [5]. Acute exacerbation of COPD (AECOPD) refers to a worsening of COPD symptoms, including increased dyspnea, cough, sputum production, and sputum purulence [6]. AECOPD is often triggered by infections, leading to a deterioration in lung function and necessitating changes in treatment, hospital admission, or intensive care [7,8]. The frequency of previous exacerbations is a strong predictor for future exacerbations, and managing and preventing exacerbations is crucial in improving a patient’s quality of life and reducing the negative impact of the disease’s progression and mortality [6,8,9,10].

Together with exacerbations of COPD, comorbidities contribute to the overall severity in individual patients and influence the disease’s progression and prognosis unfavorably [3]. An exacerbation of COPD frequently leads to hospital admission and, COPD patients tend to have prolonged hospital stays, therefore requiring a large number of medical, clinical, and financial resources [4,5,6]. When already having had an exacerbation, pronounced dyspnea and cerebrovascular insult are significantly associated with re-exacerbation [7].

The length of hospital stay (LOHS) in patients with AECOPD can be influenced by various factors, including sociodemographic [8], health-related characteristics [9], and hospital care-related features [10], as reported in previous studies. The literature shows that common factors associated with prolonged hospitalization include older age, comorbidities, and socioeconomic deprivation [8,9,10,11,12,13,14]. Older patients may experience a slower recovery, while the presence of comorbid conditions can complicate their management, both contributing to extended stays. Moreover, socioeconomically deprived patients may face barriers in accessing healthcare and post-acute care services, further prolonging hospitalization [8]. Similarly, the presence of comorbidities for example cardiovascular diseases, cancer, or diabetes, can complicate the management of AECOPD and necessitate more comprehensive treatment plans, which might consequently extend LOHS. In terms of severity, AECOPD patients with more severe exacerbations necessitating intensive interventions require longer recovery periods [12,13]. Additionally, patients with a history of frequent hospitalizations for AECOPD may have a more advanced disease and an increased susceptibility to recurrent exacerbations, necessitating longer hospital stays for comprehensive treatment and recovery [9,14]. The availability of specialized respiratory care units, access to early interventions, such as non-invasive ventilation, and timely access to medical resources can impact the course of treatment and potentially shorten a hospital stay [10]. Moreover, the availability and coordination of post-discharge care services, such as pulmonary rehabilitation or home care, are crucial factors affecting the hospital stay duration. Efficient discharge planning and access to post-discharge care services can help reduce hospital stays [14]. 

However, due to the wide range of influencing factors, there is currently no standardized approach for predicting LOHS in patients with AECOPD. The primary objective of this study was to identify factors that might predict LOHS in patients admitted for AECOPD. By identifying patient characteristics that influence LOHS, decision makers can better plan hospital management strategies. Specifically, we retrospectively investigated whether commonly available sociodemographic and health-related variables measured at admission were associated with the primary outcome LOHS. The study also aimed to establish a prediction model for the length of stay for patients hospitalized with AECOPD. While reducing LOHS may decrease hospital costs, it could also negatively affect the quality of care. Therefore, we also examined factors associated with all-cause mortality within one year and rehospitalization within 6 months as secondary outcomes. Early predictions for LOHS using variables measured at admission could potentially improve patient management, facilitate optimal discharge planning, shorten LOHS, and ultimately reduce hospital costs.

## 2. Materials and Methods

### 2.1. Study Design

This retrospective observational study was performed using existing data from patients presenting with AECOPD exacerbations at the emergency departments of the Cantonal Hospital of Baselland (KSBL), a district university teaching hospital covering a stable population of a quarter million in Northwest Switzerland [15]. We undertook a retrospective cohort study between January 2019 and February 2020. A total of 246 patients were identified who presented at the emergency departments with acute exacerbation of chronic obstructive pulmonary disease (AECOPD) 

### 2.2. Inclusion and Exclusion Criteria

Data from the electronic records of these patients were individually reviewed. Cases were included if the patients were older than 18 years, were admitted via the emergency room (ER), did not have a documented refusal of the hospital‘s general consent, and had a confirmed diagnosis of COPD exacerbation as their main diagnosis. For more information see “Informed Consent Statement”

After the application of the eligibility criteria 170 patients were included in the analysis (Figure 1).

### 2.3. Data Collection

The primary outcome of interest was LOHS. Additionally, secondary outcomes included all-cause mortality within one year and rehospitalization within six months. To minimize the risk of bias, optimism, and overfitting, we did not perform data-driven selection of variables. Instead, potential predictors were selected based on the existing literature and clinical knowledge. Two researchers conducted a comprehensive literature review and consulted clinical experts in the field. Predictors for LOHS included variables available at the time of admission: demographic variables (age, gender), vital signs (indication for oxygen supplementation), comorbidities (cardiovascular diseases, active cancer, asthma overlapping, diabetes), COPD severity (GOLD status I, II, III, IV), laboratory data (CRP, leukocytosis, eosinophil count) and admission urgency (ambulance transportation). The variable indication for oxygen supplementation was defined as both cases where the oxygen saturation was below 90% and cases where oxygen supplementation was given upon admission. For the analysis of the rehospitalization rate and mortality, events occurring during the hospitalization were also included, such as LOHS and discharge destination (rehabilitation).

The study data were collected manually, searching in all the available electronic patient records and managed using REDCap electronic data capture tools hosted at the Cantonal Hospital of Baselland [16]. The collection was performed by a doctor and a random sample of the data was checked by a health scientist.

### 2.4. Data Analysis

The primary analysis of LOHS was performed on a group of patients that were discharged alive, a sensitivity analysis was performed on the full data set. For descriptive statistics, different measures of central tendency were used based on the distribution of the data: mean and standard deviation (SD) were displayed for normally distributed variables, while the median and interquartile range (IQR) were reported for skewed variables, which were identified through histogram analysis. Analyzing age as a continuous variable allows us to capture the linear relationship between age and mortality risk, which can provide valuable insights into the gradual increase in mortality risk with advancing age (see Appendix A). For categorical variables, we presented absolute and relative frequencies. Variables with missing values were imputed using the k-Nearest neighbor algorithm (function knn.impute from the R-package “bnstruct”) [17]. A zero-truncated negative binomial regression was conducted to estimate the LOHS and its association with potential risk factors using the Rpackage “VGAM”. 

Logistic regression models were created to estimate the risk of death and rehospitalization, and its association with potential risk factors. As a sensitivity analysis, all regression models were additionally performed on the original, non-imputed data set. All statistical analyses were performed using R, version 4.0.3 statistical software [18]. All *p*-values reported were 2-sided; statistical significance was defined as *p*  <  0.05.

## 3. Results

### 3.1. Patient Characteristics 

A total of 246 cases were identified. After the application of inclusion and exclusion criteria, 170 patients were included in the analysis (Figure 1). Table 1 presents the patient characteristics of 170 patients, along with the percentage of missing data. 

The median age at admission was 75 years (range 68–79) and less than half of the patients were female (46.5%). Vital signs measured at admission revealed that more than half of the patients (51.2%) either received oxygen supplementation or had an indication for it, but only 6.5% required ventilation during hospitalization. The majority of patients (78.2%) had cardiovascular comorbidities, with 55.3% having hypertension, 26.5% having ischemic heart disease, and 19.4% arrythmias. Other comorbidities occurring most frequently were diabetes (20.6%), pneumonia (18.8%), and asthma (14.1%). Regarding the risk factors, almost half of the patients were current smokers (46.1%), and the other half (51.9%) were former smokers, whereas less than 2% were lifelong non-smokers. The median body mass index (BMI) was 23.4 with an IQR of 20.9–29.7. In terms of COPD severity based on the Global Initiative for Chronic Obstructive Lung Disease (GOLD) criteria, the majority of patients belonged to the GOLD III (35.6%) or GOLD IV class (27.9%). The laboratory values showed that almost 20% of the patients had elevated C-reactive protein (CRP) levels and 45% had leukocytosis. The median eosinophiles count was 0.06 × 10^9^/L with an IQR of 0.01–0.20 × 10^9^/L. Regarding discharge destination, circa 22% of the patients were discharged for rehabilitation. Out of the 170 hospitalized patients with AECOPD, six patients died during their hospital stay and were excluded from the regression analyses of LOHS and rehospitalization. Patients with COPD who did not die within the hospital had a median LOHS of 8 nights, with an IQR of 6–11 nights. In total, 26 patients died within one year (15.3% of the total patients) and rehospitalization at the same hospital within 6 months after discharge occurred in 44.7% of the cases.

### 3.2. Prediction of LOHS, Mortality and Rehospitalization

Multivariable zero-truncated negative binomial regression was conducted to identify predictors for the length of a hospital stay. The variables entered into the model included age, sex, smoking status, disease severity classification (GOLD stage), comorbidities (cardiovascular diseases, osteoporosis, active cancer, asthma overlapping, pneumonia, diabetes, metabolic disease), vital signs (indication for oxygen supplementation, respiratory insufficiency), laboratory values (CRP, leukocytosis, eosinophiles), and admission urgency (ambulance transportation). Table 2 provides coefficient estimates for the predictors of LOHS in patients who did not die during hospitalization. The regression coefficients are shown as an incident risk ratio (IRR). The model showed that an indication for oxygen supplementation was the only significant predictor for the length of a hospital stay (IRR = 1.281, CI = 1.097–1.496, *p*-value = 0.002). Age, sex, other comorbidities, and laboratory values did not significantly predict the length of a hospital stay in our cohort. The LOHS prediction at the intercept (8.125 days) was the LOHS when all the covariates were at 0 (for categorical covariates) or at their mean (for continuous covariates). The predicted LOHS of the model for each variable is presented for one unit increase. Patients who had an indication for oxygen supplementation at admission compared to those not requiring oxygen were predicted to remain hospitalized two nights longer (assuming all the other variables were held at a constant), the predicted LOHS rose to approximately 10 days. 

Our secondary objectives encompassed the examination of factors linked to mortality and rehospitalization rates. The outcomes of the multivariable logistic regression models for mortality are presented in Table 3. Age, severity of COPD, comorbidities, and rehabilitation exhibited significant associations with mortality in patients with AECOPD. Specifically, age (OR = 1.105 CI = 1.03–1.194, *p*-value 0.007), GOLD III or IV (OR = 4.567, CI = 1.357–19.415, *p*-value 0.023), and active cancer (OR = 7.954, CI = 2.073–32.749, *p*-value 0.003) were positively associated with an increased risk of death within 12 months of admission. On the other hand, rehabilitation after discharge was negatively associated with mortality (OR = 0.071, CI = 0.002–0.609, *p*-value 0.049) meaning that people who underwent rehabilitation had higher chances to survive a year after the studied admission. No other variable demonstrated a statistically significant association with 1 year mortality.

Table 4 presents the findings of our secondary multivariable analysis regarding the rate of rehospitalization within 6 months. The logistic regression analysis revealed that none of the variables included in the model showed any statistically significant associations with rehospitalization.

## 4. Discussion

Our retrospective cohort study is a unique recent investigation conducted in Switzerland, focusing on real-world data, with the aim of identifying possible predictors for LOHS, mortality, and rehospitalization in patients with AECOPD. The main findings are that an indication for oxygen supplementation at admission was a significant predictor of LOHS, while older age, COPD GOLD III and IV, and active cancer were significantly associated with higher mortality, and rehabilitation after discharge was associated with lower mortality. The logistic regression analysis revealed that none of the variables included in the model showed any statistically significant associations with rehospitalization. 

The demographic characteristics and outcome variables of our study sample were similar in comparison to other studies [19,20,21]. Our analysis included 170 patients with a median age of 75 years with a predominance of males in the study population. COPD used to be a condition that mainly affected men. However, in recent years, there has been a significant worldwide increase in the prevalence of COPD among women. As a consequence global mortality and hospitalization rates related to COPD have increased [22,23,24]. Data published in the Swiss COPD cohort (timeframe 2005–2014) showed that from a total cohort of 1312 individuals 39.6% were females, whereas five years later, in our study, the proportion rose to 46.5% [2]. Given the well-established association between smoking and the development of COPD [25] it was expected that our study population predominantly comprised current or former smokers, accounting for 98% of the sample. Similarly, in terms of comorbidities, our study sample is representative of the overall COPD population, because 84% of the patients had at least one chronic condition, which is consistent with the previous findings where the proportion was around 80% [26]. Overall the LOHS in patients with COPD has declined significantly in Switzerland over the past 15 years [27]. The average LOHS in our study population (8 days) is in line with the national trend. Moreover, the European COPD Audit in 13 countries, which included data from 16018 hospitalized patients revealed that people with COPD had an average LOHS of 7 days [13].

### 4.1. Predictors of LOHS, Mortality and Rehospitalization

As previously mentioned, it was observed in our research that an indication for oxygen supplementation at admission was the only significant predictor of the LOHS. The predicted value of a hospital stay increased by 2 days (from 8.1 days to 10.3 days) when patients needed oxygen supplementation or their oxygen saturation on admission was <90%. The need for supplemental oxygen at admission reflects the oxygenation status and the severity of the airflow obstruction and may be an indicator of increased complications. Even though oxygen supplementation has to be applied carefully in COPD with a target oxygen saturation of 88–92% due to the elevated risk of hypercapnia, GOLD and other respiratory societies consider it a key component in the management of severe COPD exacerbation due to its positive effects on patient outcomes [28,29,30]. The observed significance of an indication for oxygen supplementation as a predictor for the length of a hospital stay is in line with previous research studies, including observational and RCTs [13,31,32,33].

Our secondary aim was to evaluate the factors associated with mortality in AECOPD. The overall in-hospital morality (3.5%) was lower than that reported in the European audit of COPD in 2013 (4.9%) [34]. The study conducted in Switzerland and published by Kutz, et al. in 2019, however, identified a trend where the overall risk-adjusted all-cause in-hospital mortality declined from 4.9% in 2009 to 4.6% in 2015 [27]. The analysis of mortality within one year showed that older age, severe airflow limitation, active cancer, and arrhythmias were positively associated with mortality, and interestingly, rehabilitation was negatively associated. Age is a known significant factor associated with mortality across various populations and disease conditions. Therefore, it is not surprising that our study findings also revealed a positive association between age and mortality in individuals with COPD. Numerous studies conducted in different settings have demonstrated that advancing age is associated with increased mortality rates, specifically within the COPD population [34,35,36,37,38]. The underlying mechanisms contributing to this association are multifactorial and complex. Aging is accompanied by physiological changes, including a decline in organ function, compromised immune response, and an increased susceptibility to comorbidities. These age-related factors can lead to increased complications and contribute to adverse outcomes, ultimately leading to higher mortality rates. Healthcare providers should therefore identify older individuals with COPD who may be at higher risk of mortality and tailor their treatment plans accordingly. Close monitoring, early intervention during exacerbations, and comprehensive management of comorbidities are crucial for optimizing outcomes in this vulnerable population. The Global Initiative for Chronic Obstructive Lung Disease (GOLD) severity stage classifies COPD into four stages based on airflow limitation (measured by forced expiratory volume in one second FEV1) [28]. In accordance with other studies [34,39,40,41,42], we also observed that mortality increased with the severity of airflow limitation, defined by the GOLD stages. Particularly, COPD patients classified in the severity GOLD stage III or IV had an increased risk of mortality compared to the mild or moderate stage. The alteration in GOLD stages holds promise as a monitoring tool and outcome measure for clinical research. Although the prognostic value of GOLD status is widely known and verified, it is important to note that in our study this value was reported in only 61.2% of cases. Since 2017, GOLD has proposed a symptom- and exacerbation-based categorization system consisting of groups A to D. [43] However, despite our dataset being sourced from 2019, we found that only 37.6% of the patients in our study cohort had documented information regarding their classification into GOLD groups A–D. As a result, we were unable to incorporate this classification system into our analysis.

Another point worth discussing is the role of comorbidities on mortality. It is already known from the literature that comorbidities are common among people with COPD and they significantly impact patients’ survival [44,45,46]. A strong finding in our research is that arrhythmias emerged as the only significant cardiovascular comorbidity associated with mortality, and the most common reason for arrhythmias in our sample was atrial fibrillation (AF). AF is a common type of arrhythmia characterized by irregular heart rhythms originating from the atria. It can lead to poor blood flow, thrombosis, and other cardiovascular complications [47,48]. The association between AF and mortality in patients with COPD exacerbation is consistent with previous research findings [49,50,51]. These studies have demonstrated that the presence of AF is an independent predictor for adverse outcomes, including increased mortality among patients with COPD. The identification of arrhythmias, particularly AF, as a significant comorbidity associated with mortality highlights the importance of considering cardiac complications in COPD patients. The other significant comorbidity was active cancer. The role of cancer in all-cause mortality aligns with the existing evidence and supports the notion that active cancers are a well-established factor associated with increased mortality [52,53,54]. Moreover, it is worth mentioning that some comorbidities are also associated with tobacco usage, which is the main cause of COPD, including coronary heart disease, congestive heart failure, and especially lung cancer [55,56,57,58]. However, except for lung cancer these comorbidities did not show a significant association with mortality in our study. Finally in order to control for the potential confounding effect of pneumonia we conducted a sensitivity analysis. The results of the models aligns with the primary models, providing additional evidence of the robustness and consistency of our findings (the sensitivity analysis results are displayed in the Appendix A). 

Interestingly, our results suggest that rehabilitation after hospitalization for AECOPD is associated with improved outcomes and a reduced mortality. This highlights the importance of incorporating rehabilitation interventions into the comprehensive management of patients with AECOPD. Particularly pulmonary rehabilitation programs implemented after an acute exacerbation of COPD have been confirmed to be associated with a significant reduction in mortality [34,59,60,61]. These findings support the current guideline recommendations for pulmonary rehabilitation after hospitalization for AECOPD. A notable modification in the GOLD 2023 report is the increased focus on reducing mortality as a primary objective of treatment, which is reflected in the inclusion of a novel segment titled “Therapeutic interventions to decrease COPD mortality” [62]. This report highlights the role of non-pharmacological interventions, such as pulmonary rehabilitation in reducing mortality. 

Concerning our secondary analysis, assessing the association of parameters with rehospitalization within six months, our findings highlight the unpredictability of rehospitalization based on demographic factors, comorbidities, and laboratory values at admission in patients with AECOPD and emphasize the need for comprehensive variable inclusion in future studies. Variables that indicate the course of the disease, such as exacerbation history and symptom severity (e.g., GOLD ABCD classification), should be considered in the development of future prediction models. Incorporating these factors into the predictive model may enhance its accuracy and provide clinicians with valuable insights for identifying patients at a higher risk of rehospitalization.

### 4.2. Strengths, Limitations, and Further Research

The novelty of our study is that our analysis expanded its focus to encompass a range of factors, including demographic factors, health-related factors, and laboratory values available at the time of admission. A further strength was being able to investigate long term outcomes, such as mortality within one year as this data was available for all patients. Moreover, all statistical analyses that were primarily performed on the imputed data were also applied to the non-imputed dataset, the statistical significance of the variables with missing data did not differ in the two datasets. The sensitivity analysis demonstrated that the internal validity of our research was not impacted by the missing data. There are some limitations to this study. Being a retrospective study, the data quality depends on precise documentation in the patient files. The data collection process might have resulted in incomplete documentation, leading to missing information, particularly details about smoking, including the number of cigarettes smoked daily and cumulative exposure measured in pack-years, which are critical factors in the progression and prognosis of COPD. Moreover, we assumed that if the GOLD status was not reported, the clinician did not assess it, so we could have underestimated the percentage of patients with a GOLD status classification upon hospital admission. Additionally, information about rehospitalization within 6 months was only possible within KSBL; due to the privacy policy it was not possible to access information about rehospitalizations in other hospitals. However, in Switzerland readmissions usually occur within the same hospital; a Swiss study has shown that only 17% of unplanned readmissions occurred at a different hospital [41].

General practitioners (GPs) have a unique perspective on COPD management as they provide long-term care and monitoring to patients outside of the hospital setting. They possess valuable insights into the day-to-day fluctuations in symptoms, adherence to treatment regimens, and the effectiveness of interventions implemented in primary care. The combination of GP and hospital data can potentially enhance the accuracy and reliability of predictive models for COPD rehospitalization. By incorporating variables, such as exacerbation frequency, medication adherence, smoking status, and GP intervention strategies, researchers can develop robust risk prediction models that encompass both prehospitalization and post-discharge periods. Future studies incorporating data obtained from GPs regarding the progression of COPD with hospital records can potentially enhance the accuracy and reliability of predictive models for COPD hospitalization and rehospitalization.

Despite these limitations, our study provides a foundation for future research and contributes with valuable insights into other aspects of COPD, particularly focusing on the possible predictors of LOHS, mortality, and rehospitalization, available at admission time. The identification of predictors available at admission time might help to promptly identify patients at higher risk of adverse outcomes and consequently healthcare providers can prioritize their care, allocate appropriate resources, and develop personalized management strategies tailored to their specific needs. This proactive approach ensures that patients receive timely and targeted interventions, potentially leading to improved outcomes and a reduced burden on healthcare systems. Furthermore, the ability to identify predictors at admission time facilitates the development of predictive models or risk stratification tools. These tools can assist healthcare professionals in making informed decisions about the level of care required, resource allocation, and appropriate follow-up strategies. By incorporating these predictors into clinical practice, healthcare systems can optimize resource utilization, enhance patient triage, and improve overall healthcare delivery.

## 5. Conclusions

This study identified several predictors for LOHS and mortality, which may further advance our understanding of AECOPD and thereby improve patient management, discharge planning, and hospital costs. In fact, indication for oxygen supplementation at admission was a significant predictor of LOHS, whereas older age, severe airflow limitation and comorbidities were significantly associated with higher mortality, while rehabilitation was associated with lower mortality. The logistic regression analysis revealed that none of the variables included in the model showed any statistically significant associations with rehospitalization, emphasizing the need for future, larger studies with a more comprehensive variable inclusion. Variables that indicate the course of the disease, such as exacerbation history and symptom severity, should be considered in the development of future prediction models. Interestingly, rehabilitation after hospitalization was associated with lower mortality. This potentially protective effect of rehabilitation after hospitalization clearly warrants further confirmation and validation because, due to the non-randomized design of our study, causal inference cannot be drawn. However, rehabilitation has high potential to further improve the comprehensive management of patients with AECOPD.

## Figures and Tables

**Figure 1 jcm-12-05322-f001:**
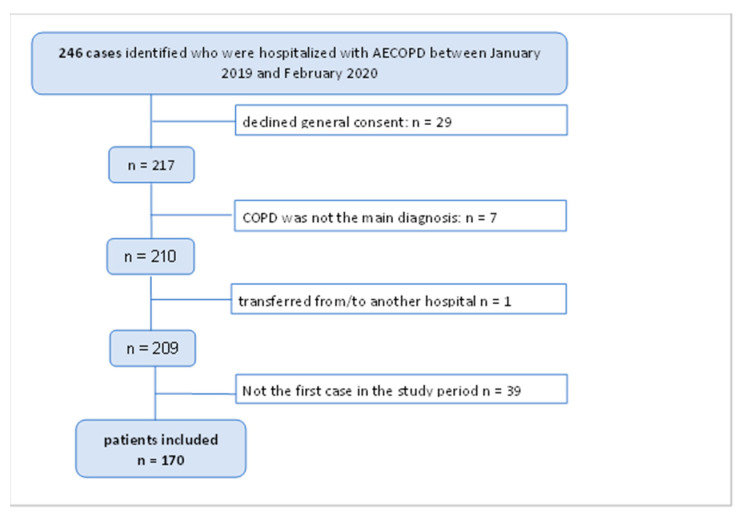
Flowchart diagram for patient selection process.

**Table 1 jcm-12-05322-t001:** Patient characteristics.

	All (n = 170)	Missing (%)
**Demographic**		
Age at diagnosis, median [IQR] in years Age at diagnosis, median (IQR) in years	75 (68–79) 73 (62–81.5)	--
Gender (Male), n (%)	91 (53.5%)	--
**Vital Signs**		
Indication for oxygen supplementation, n (%)	87 (51.2%)	--
Respiration insufficiency, n (%)	18 (16.51%)	
Ventilation during hospitalization	11 (6.5%)	
**Comorbidities**	169 (99.4%)	
Cardiovascular diseases	133 (78.2%)	--
Heart failure, n (%)	12 (7.1%)	--
Ischemic heart disease, n (%)	45 (26.5%)	--
Arrhythmias, n (%)	33 (19.4%)	--
Peripheral artery disease, (PAD), n (%)	25 (14.7%)	--
Hypertension, n (%)	94 (55.3%)	--
Osteoporosis, n (%)	18 (10.6%)	--
Cancer, n (%)	17 (10.0%)	
Asthma-COPD overlap, n (%)	24 (14.1%)	--
Pneumonia, n (%	32 (18.8%)	--
Diabetes, n (%)	35 (20.6%)	--
Metabolic syndrome, n (%)	19 (11.2%)	--
**Risk Factors**		
Smoking status		(9.4)
Current smoker, n (%)	71 (46.1%)	
Former smoker, n (%)	80 (51.9%)	
Never smoker, n (%)	3 (1.9%)	
BMI, median [IQR]	23.4 [20.9, 29.7]	(27.1)
**COPD severity stage**		(38.8)
GOLD I, n (%)	7 (6.7%)	--
GOLD II, n (%)	31 (29.8%)	--
GOLD III, n (%)	37 (35.6%)	--
GOLD IV, n (%)	29 (27.9%)	--
**Laboratory Values**		
CRP value (mg/L)	29.00 [7.25, 63.75]	(0.6)
CRP elevated *, n (%)	33 (19.5%)	(0.6)
Leucocytosis **, n (%)	76 (45%)	(0.6)
Eosinophil count *** (10^9^/L)	0.06 [0.01, 0.20]	(5.3)
**Admission and discharge circumstances**		
Transportation upon admission: ambulance, n (%)	29 (17.1%)	--
Discharge destination: rehabilitation, n (%)	38 (22.4%)	
**Outcomes**		
Length of stay, in nights, median [IQR]	8 (6, 11)	--
Rehospitalization within 6 months, n (%)	76 (44.7%)	--
Death		--
Death (in hospital death)	6 (3.5%)	--
Death (one year mortality)	26 (15.3%)	--

* CRP ≥ 5 mg/L. ** Leucocytes elevated > 10.5 × 10^9^/L. *** Eosinophil count normal range < 500 × 10^9^/L).

**Table 2 jcm-12-05322-t002:** Results of multivariable zero-truncated negative binomial regression model for length of hospital stay (LOHS) estimation in AECOPD survivors (*n* = 164).

	LOHS Prediction	IRR	(95%CI)		*p*-Value
(Intercept)	8.125	5.647	2.83	11.267	0.000
Sex	7.550	0.928	0.789	1.092	0.368
Age	8.167	1.005	0.996	1.014	0.242
Severe COPD (GOLD III or IV)	7.718	0.949	0.805	1.119	0.534
Heart failure	7.593	0.934	0.699	1.248	0.642
Ischemic heart disease	7.090	0.871	0.731	1.037	0.119
Arrhythmias	7.600	0.934	0.761	1.147	0.516
Peripheral artery disease (PAD)	9.106	1.122	0.91	1.384	0.281
Diabetes	7.490	0.921	0.754	1.124	0.417
Asthma-COPD overlap	8.424	1.037	0.83	1.296	0.747
Active cancer	6.462	0.792	0.608	1.031	0.083
Ambulance transportation	8.830	1.088	0.925	1.28	0.310
**Indication for oxygen supplementation**	**10.386**	**1.281**	**1.097**	**1.496**	**0.002**
CRP	8.128	1.000	0.999	1.001	0.494
Leucocytosis	7.240	0.889	0.761	1.04	0.142
Eosinophils	5.485	0.667	0.402	1.107	0.118

**Table 3 jcm-12-05322-t003:** Results of multivariable logistic regression model for one year mortality in patients with COPD.

	OR	(95%CI)		*p*-Value
Sex	0.481	0.141	1.547	0.227
**Age**	**1.105**	**1.03**	**1.194**	**0.007**
**Severe COPD (GOLD III or IV)**	**4.567**	**1.357**	**19.415**	**0.023**
Heart failure	0.809	0.089	4.586	0.826
Ischemic heart disease	0.624	0.161	2.108	0.465
Arrhythmias	**7.686**	**2.399**	**27.695**	**0.001**
Peripheral artery disease (PAD)	0.759	0.152	3.08	0.714
Diabetes	1.929	0.541	6.662	0.299
Asthma-COPD overlap	0.644	0.077	3.42	0.637
**Active cancer**	**7.954**	**2.073**	**32.749**	**0.003**
**Rehabilitation**	**0.071**	**0.002**	**0.609**	**0.049**
Length of hospital stay	0.932	0.811	1.061	0.299

**Table 4 jcm-12-05322-t004:** Results of multivariable logistic regression model for one year mortality in patients with COPD.

	OR	(95%CI)		*p*-Value
Sex	1.612	0.798	3.284	0.184
Age	1.027	0.989	1.068	0.169
Severe COPD (GOLD III or IV)	0.552	0.27	1.115	0.1
Heart failure	1.446	0.416	5.331	0.562
Ischemic heart disease	1.248	0.59	2.631	0.56
Arrhythmias	0.777	0.325	1.814	0.562
Peripheral artery disease (PAD)	1.735	0.685	4.514	0.248
Diabetes	1.353	0.586	3.157	0.479
Asthma-COPD overlap	1.415	0.529	3.77	0.485
Active cancer	2.468	0.845	7.887	0.107
Rehabilitation	1.22	0.506	2.984	0.658
Length of hospital stay	0.966	0.888	1.047	0.407

## Data Availability

All data generated were analyzed during this study and the results included in this article. The data presented in this study are available on reasonable request from the corresponding author. The data are not publicly available due to restrictions in data privacy.

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
