# Peer review of "Predictors of Length of Stay, Mortality and Rehospitalization in COPD Patients: A Retrospective Cohort Study"

_jcm, 2023, doi:10.3390/jcm12165322_

Round 1

Reviewer 1 Report

The manuscript "Predictors of length of stay, mortality and rehospitalization in COPD patients: a retrospective cohort study " is interesting as it focuses on the Acute Exacerbation of Chronic Obstructive Pulmonary Disease (AECOPD). AECOPD is a severe occurrence that results in sudden deterioration of breathing symptoms in people with COPD. It presents significant challenges and can cause severe complications that increase mortality rates. These complications include respiratory distress, reduced lung function, higher risk of respiratory infections, hospitalization, and accelerated decline in lung function. Additionally, managing AECOPD requires complex treatment, and hospitalization due to AECOPD can burden healthcare systems substantially.

COPD is a prevalent chronic lung disease that significantly impacts individuals and healthcare systems worldwide. This study analyzed data from 170 COPD patients at a Swiss hospital to predict factors affecting the length of hospital stay, one-year mortality, and rehospitalization after the AECOPD. 

It would be helpful if the authors could provide more detail on the factors influencing the length of hospital stay for AECOPD, specifically in lines 47-50. The current information on these factors is quite brief and general.

Could You also provide in Table 1 the range of age for males and females separately?

The authors have performed the multivariable zero-truncated negative binomial and logistic regression analysis. The description of the results lacks some additional explanation, and it should be discussed more in detail to attract the broad scientific community’s interest. For example, two factors that increase the risk of mortality in one year after AECOPD are age and active cancer.

Many studies have identified older age as a predictor of increased mortality risk for those with AECOPD.  The older the patient, the higher the risk of mortality during an exacerbation. Various factors contribute to higher mortality in AECOPD patients with older age, such as reduced physiological reserves, comorbidities, reduced immune response, frailty, and multiple medication use. It would be beneficial for the readers If You could discuss whether there is a specific age threshold beyond which mortality increases or if certain age groups are more prone to mortality due to AECOPD or additional age-related factors.

The second factor associated with post-AECOPD mortality is the presence of cancer – there are no details of what kind of cancer was present among the studied cohort. I am aware that patients may have various cancer types or undergo different treatments, but more details should be added.

Also, readers should be given more details on rehabilitation as the protective factor leading to decreased mortality. What kind of rehabilitation do patients obtain? Was there any specific minimal length of the rehabilitation considered in your analysis?

Another question is the correlation between smoking, obesity, with length of stay, post-AECOPD mortality, or rehospitalization was researched. Smoking and obesity, present as risk factors in Table 1, and mentioned as considered in the multivariate analysis – are not presented in the following paragraphs and Tables. Were smoking status, metabolic disease, and obesity (or BMI) included in the multivariable regression analysis? If so, please include the data in Tables 2,3 and 4.

Lastly, did you also perform the multivariate analysis? 

Reviewer 2 Report

In this retrospective study the authors identify the factors that predict the lenght of hospital stay, rehospitalization and one-years mortality in patients with acute exacerbation of COPD. In particular, the author conclude that indication for oxygen therapy were associated with a predictor of lenght of hospital stay, sever airflow limitation and comorbidities were associated with higher mortality while rehabilitation was associated with lower mortality.

I suggest to the autors the following changes to improve the quality of manuscript:

-In the introduction the authors should insert the definition of acute exacerbation of COPD

-Patients with pneumonia shoul be excluded from analysis

 Data analysis is correct Experimental design is appropriated to achieve the aim of the study  and the result are interesting.

English language shoul be improved

Reviewer 3 Report

The authors have conducted a retrospective cohort study to identify the predictors of LOHS, mortality and rehospitalization in COPD patients.

It is a well planned and executed study. 

There are no major revisions needed.

A few minor questions are as follows:

1. What questionnaire was used to document general consent? and If the refusal was not documented what was the course of action?

2. the line 77 pertaining to the above question may be rewritten to prevent misunderstanding if informed consent was received and documented.

3.line 124 states 264 cases but elsewhere the number is mentioned as 246 is it a typo?
